# Pyrimidine Biosynthetic Enzyme CAD: Its Function, Regulation, and Diagnostic Potential

**DOI:** 10.3390/ijms221910253

**Published:** 2021-09-23

**Authors:** Guanya Li, Dunhui Li, Tao Wang, Shanping He

**Affiliations:** 1State Key Laboratory of Developmental Biology of Freshwater Fish, Hunan Provincial Key Laboratory of Animal Intestinal Function and Regulation, Hunan International Joint Laboratory of Animal Intestinal Ecology and Health, Hunan Normal University, Changsha 410081, China; LGY_1228@163.com; 2Colllege of Nursing and Health, Zhengzhou University, Zhengzhou 450001, China; dunhui.li@murdoch.edu.au; 3Centre for Molecular Medicine and Innovative Therapeutics, Murdoch University, Perth 6150, Australia; 4Perron Institute for Neurological and Translational Science, the University of Western Australia, Nedlands 6009, Australia

**Keywords:** carbamoyl phosphate synthetase, aspartate transcarbamoylase, dihydroorotase, regulation, pyrimidine, diseases, cancer

## Abstract

CAD (Carbamoyl-phosphate synthetase 2, Aspartate transcarbamoylase, and Dihydroorotase) is a multifunctional protein that participates in the initial three speed-limiting steps of pyrimidine nucleotide synthesis. Over the past two decades, extensive investigations have been conducted to unmask CAD as a central player for the synthesis of nucleic acids, active intermediates, and cell membranes. Meanwhile, the important role of CAD in various physiopathological processes has also been emphasized. Deregulation of CAD-related pathways or CAD mutations cause cancer, neurological disorders, and inherited metabolic diseases. Here, we review the structure, function, and regulation of CAD in mammalian physiology as well as human diseases, and provide insights into the potential to target CAD in future clinical applications.

## 1. Introduction

The route of formation of carbamoyl phosphate (CAP) was first discovered in microorganisms in 1955 [1]. Later, genetic studies revealed the role of enzymatic synthesis of CAP in the pyrimidine pathway in Neurospora [2,3]. It was also found that carbamyl phosphate synthetases (CPSases) CPS-1 and CPS-2 provide CAP pools for arginine and pyrimidine synthesis, respectively [4,5,6]. In animals, aspartate transcarbamoylase (ATC) and dihydroorotase (DHO) were subsequently co-purified with CPS-2. These three enzymes form a single multi-enzymatic protein named CAD to participate in the de novo pyrimidine pathway in mammals [7,8].

Over the past two decades, studies from dozens of labs have revealed that CAD takes part in the de novo pyrimidine nucleotide synthesis, and plays a leading role in cellular and organismal physiology in various forms of life [9,10,11,12,13]. Moreover, CAD is also involved in protein glycosylation and biosynthesis of phospholipids [14]. In recent years, extensive studies have been carried out on the molecular functions and pathways of CAD to address the unanswered questions in this area. In this review, we provide an overview of the current understanding of CAD and its critical roles in metabolism, physiological regulation, as well as disease progression.

## 2. CAD Structure and Function

### 2.1. Overall Structure and Function of CAD

CAD is a multifunctional protein that takes part in the initial three speed-limiting steps of pyrimidine nucleotide synthesis. Moreno-Morcillo et al. have shown that CAD is a hexamer of a 243 kDa polypeptide chain [15]. Human CAD involves the concerted action of four domains: glutamine amidotransferase (GATase), carbamylphosphatesynthetase II (CPSIIase), aspartate transcarbamylase (ATCase), and dihydroorotase (DHOase) (Figure 1A). CPSIIase consists of two highly homologous fragments, which are designated as CPSase A and CPSase B. Specifically, GATase and CPSase (CPSase A and CPSase B) jointly form the glutamine-dependent CPSase. GATase transfer HCO_3_^−^, glutamine, and ATP to form carbamoylphosphate (CP) at the site of CPSIIase domain. CP formation is the first rate-limiting step for the nucleotide synthesis [16,17]. We will discuss the details of CPSIIase and its regulations in the latter text. ATCase, consisted of a catalytic homotrimer, catalyzes carbamoylphosphate (CP) and aspartate (Asp) into carbamoyl aspartate (CA-asp) [18]. The DHOase domain catalyzes the reversible cyclization of CA-asp to dihydroorotate (DHO), the first cyclic compound of de novo pyrimidine nucleotide synthesis pathway [15,19,20]. Dihydroorotate dehydrogenase (DHODH) subsequently reduces DHO to orotate in mitochondria [21]. Finally, uridine monophosphate synthase (UMPS) converts orotic acid to produce the end product uridine monophosphate (UMP) [22,23,24] (Figure 1B).

Structural studies of CAD have provided significant insights into its internal structural organization and function. The crystal structures of CAD and its fragments have been determined, and CAD is evolutionarily conserved in a variety of organisms [25,26,27]. Human CAD has a hexamer structure of 1.5 MDa, which is made of a couple of 243 kDa polypeptide chains. This particular structure endows human CAD a series of sophisticated catalytic and regulatory properties [28]. As briefly mentioned earlier, feedback inhibition and allosteric activation by UTP and PRPP play a predominant role in the overall regulation of CAD’s CPSIIase domin [17,29]. The DHOase and ATCase domains are connected by a linker to form a hexamer, which is proposed as the central structural element of CAD [15]. Interestingly, CAD, as a fusion of the first three enzymatic activities, only occurs in animals. De novo biosynthesis of pyrimidine nucleotide is mainly catalyzed by distinct or independent enzymes but not a single multi-enzymatic protein in prokaryotes [11].

### 2.2. CAD Participates in Pyrimidine Nucleotide Biochemistry and Metabolism

The de novo biosynthesis of pyrimidine nucleotides provides essential precursors for multiple growth-related events in higher eukaryotes [30,31,32]. Intracellular pyrimidine nucleotides regulate the steps of many metabolic pathways (Figure 2A,B), mainly including nucleic acid precursors, activated intermediates, and membrane synthesis. Indeed, the de novo biosynthetic pathway in mammals is capable of supplying all pyrimidine ribonucleotides and deoxyribonucleotides for RNA and DNA biosynthesis [33,34]. UMP, as an intermediate product of the synthesis of pyrimidine nucleotides, can be further dephosphorylated to uridine (UR) and phosphorylated to corresponding di- (UDP) and triphosphorylated (UTP) forms, respectively. As described in the latter, UTP is involved in protein glycosylation and glycogen synthesis with the form of UDP-linked sugars [35]. CTP synthetase (CTPS) converts UTP into CTP in an ATP-dependent reaction that uses glutamine as an amine donor. Same as UTP, CTP can also be dephosphorylated into CDP and CMP. Alternatively, UDP and CDP can be deoxygenated into deoxy-UDP (dUDP) and dCDP, respectively, by ribonucleotidereductase (RNR), and further phosphorylated by NDPK. To avoid misincorporation into DNA, dUTP is rapidly broken down by dUTPase into dUMP. Importantly, thymidylate synthase (TS) transfers UMP intodeoxy-TMP (dTMP) and is phosphorylated subsequently into dTTP. Thus, the de novo biosynthetic pathway in mammals is capable of supplying all pyrimidine ribonucleotides (CTP, UTP) and deoxyribonucleotides (dCTP, dTTP) for RNA and DNA biosynthesis, respectively [33,34,36].

At the same time, CAD plays predominant roles in the production of activated intermediates, including pyrimidine sugars, polysaccharide, and phospholipid synthesis [37]. UDP, as one of the products of CAD, is also a precursor of UDP-sugar intermediates, which are required in UDP-dependent glycosylation events and post translational modification of proteins. Besides, these UDP-sugar intermediates are potential extracellular signaling molecules. For example, UDP-N-acetylglucosamine (UDPGlcNAc) is involved in post-translational modifications of proteins and is required for UDP-dependent glycosylation of proteins. [38]. Of interest, a recent report found that the UDP-nucleotide sugar, as well as UDP-glucose pyrophosphorylase, correctly response to cancer process in pancreatic cancer and breast cancer by disrupting cancer cell glycosylation [39,40]. However, CAD mutants decrease glycosylation precursors. It has been identified that CAD mutants (CAD ^hu10125^) regulates angiogenesis in the embryo in Zebrafish via CAD mediated glycosylation process [31]. Additionally, biallelic mutations in CAD (c.1843-1G > A, c.6071G > A) also confirm the previous study, and manifest with a reduction in a subset of UDP nucleotide sugar levels (UDP-glucose, UDP-N-acetylglucosamine, UDP-galactose, etc.) that serve as donors for glycosylation in human [41]. Despite these insights, how is nucleotide sugar level regulated? Why are glycosylation levels alleviated in cancer? These questions have been elusive until very recently.

Cell membrane and synapses phospholipid mainly consist of phosphatidylcholine (PtdCho) in eukaryotic cells, and CTP participates in PtdCho synthesis in the CDP-choline pathway, which are required for cellular growth and repair, and specifically for synaptic function [42,43]. UMP, as product of the de novo pyrimidine biosynthetic pathway, up-regulates CDP-choline levels and other major membrane phosphatides by increasing CTP level [44,45]. Furthermore, pyrimidine-dependent nucleotide–lipid cofactors are required for erythrocyte membrane synthesis, and shortage of these cofactors might result in dyserythropoiesis [46].

## 3. The Regulation of CAD

Recently, the identification and characterization of CAD-interacting proteins has yielded deep insights into the regulation of CAD catalytic activity. The first identified CAD-interacting protein is the Rad9 checkpoint protein, a subunit of heterotrimeric Rad9-Rad1-Hus1 (9-1-1) complex. The role of Rad9 in predicting disease is likely important. Its overproduction is essential for tumor cells’ tumorigenicity and metastasis [47,48,49]. Additionally, Rad9 binds to CAD at its CPSIIase domain, resulting in a 2-fold stimulation of the CPSase activity of CAD and thus promoting the ribonucleotide biosynthesis [50]. It is supposed that CAD is likely a target gene of Rad9 and is controlled by transcription regulation. Additionally, Nakashima et al. have shown that mLST8, a component of the mTOR complexes, interacts with CAD and its binding with CAD is stronger than that of mTORC1. The activity of CAD is upregulated through the association with mLST8 [51]. Consistent with this, CAD activity is enhanced when it binds to Rheb, a member of the Ras superfamily small GTPases, which regulates protein synthesis and cell proliferation in an mTOR-independent pathway [52]. These data suggest that CAD interaction proteins are involved in CAD regulation.

Furthermore, posttranslational modifications (PTMs), such as phosphorylation [53,54], acetylation [55,56], methylation [57], and ubiquitination [58], modulate the activity, localization and other properties of a protein, thereby regulating a variety of biological processes, including gene transcription, protein biosynthesis, cellular signaling, and metabolism [59]. Additionally, protein acetylation plays a pivotal role in mediating protein function, stability, and localization. Functionally, N-terminal acetylationis were found to regulate protein degradation and inhibit protein translocation into the endoplasmic reticulum. It is inferred from combination of experimental and computational evidence that CAD can be potentially acetylated at alanine residue (amino acid 2), N6-acetyllysine (amino acid 747) and N6-acetyllysine (amino acid 1411), while their functions in cellular process still needs to be further demonstrated [60,61,62]. Nowadays, researches on PTMs of CAD protein mostly focus on the phosphorylation; the phosphorylation of CAD protein and the involved pathway is depicted in Figure 3. Unfortunately, less research is conducted on other PTMs of CAD.

### 3.1. MAPK/cAMP-Dependent PKA/PKC Pathway

Stimulation of dormant cells by growth factors, such as epidermal growth factor (EGF), insulin-like growth factor-I (IGF-1) and platelet-derived growth factor (PDGF), activates the mitogen-activated protein kinase (MAPK) (Erk1/2) cascade and triggers the transition to the proliferating stage [63,64]. One of the important substrates of MAPK is CAD. In resting cells, CAD stays cytosolic and keeps an unphosphorylated condition. Once cells are activated by EGF, CAD translocates to the nucleus and phosphorylation reaction occurs at CAD Thr456 by the MAPK [65]. Moreover, after sequential phosphorylations of CAD CPSII via MAPK at CAD Thr456, which occurs just before entry into the S phase of the cell cycle, CAD loses its feedback inhibition by UTP, turning more sensitive to phosphoribosyl pyrophosphate (PRPP) [16]. Protein kinase C (PKC) not only promotes phosphorylation at CAD Ser1873, but also activates Raf and MAPK, thereby promoting phosphorylation of CAD Thr456 [13]. Phosphorylation of CAD Thr456 plays important roles in cell cycle-dependent regulation of de novo pyrimidine biosynthesis [66,67,68]. However, the phosphorylation of CAD at the site of Ser1406 by cAMP-dependent protein kinase A (PKA) can decrease its sensitivity to PRPP and abolish UTP inhibition, while phosphorylation of the site of Ser1859 has a minor effect on the catalytic activities or allosteric transitions of CAD [69]. In the process, PKA-mediated phosphorylation of a distinct site on Raf can downregulate MAPK, whereas PKC activates MAPK through direct phosphorylation and activation of Raf [70]. PKC and PKA display synergistic and antagonistic interactions in MAPK-mediated cell cycle-dependent regulation of pyrimidine biosynthesis to ensure that up-and down-regulated signals do not conflict [71].

### 3.2. PI3K-AKT-mTORC1-S6K1 Pathway

The mechanistic target of rapamycin (mTOR) is a serine/threonine protein kinase, which is usually assembled into two complexes, including mTOR complex 1 (mTORC1) and 2 (mTORC2) [72]. mTOR senses and integrates growth signals to regulate cell growth, proliferation, and homeostasis [53,73,74,75], mostly by regulating several anabolic and catabolic processes [76,77]. Physically, mTORC1 stays in the cytoplasm, while it turns to be located on the lysosomal surface via amino acids stimulation and is activated by Rheb, a small GTPase, which is localized on the lysosomal membrane. mTORC1 phosphorylates its downstream target ribosomal protein S6 kinase 1 (S6K1) and subsequently directly phosphorylates CAD Ser1859 and Ser1900 [52,78,79,80]. Interestingly, CAD accumulates on lysosomes with Rheb in a GTP- and effector domain-dependent manner [52]. Additionally, insulin also stimulates the Ser1859 phosphorylation of CAD in a manner sensitive to rapamycin as well as phosphatase and tensin homolog (PTEN) deletion in the PI3K–AKT pathway [53,81,82]. That is, CAD can be regulated both by amino acids and growth factors. Recent studies have suggested that once amino acids are taken up into cells and accumulated on lysosomes, CAD might associate with mLST8 and Rheb [51,72]. Thus, CAD localization to lysosomes may provide a physical environment and give itself better access to glutamine, which is needed for de novo pyrimidine biosynthesis [83].

## 4. Implications for Therapy of CAD-Related Diseases

The production of pyrimidine nucleotide is constantly changing to meet the pyrimidine nucleotide needs of cells. Some conditions, such as the microenvironment of tissues, hormonal context, and nucleic acid integrity, may disturb the homeostasis of pyrimidine nucleotide metabolism [84,85]. Upregulation of de novo pyrimidine synthesis plays pivotal roles in human diseases, which highlights the importance of its diagnostic potential in associated clinical settings and contributes to the elucidation of the development of novel therapeutic strategies.

### 4.1. CAD and Tumors

Metabolic reprogramming mostly indicates the proliferation and survival of cancer cells [82]. Numerous findings have strong implications for cancer therapeutic strategies. Wang et al. found pyrimidine metabolism signaling pathway was disrupted and CAD was enriched in a set of cancer types (liver cancer, breast cancer, colon cancer, etc.) with poor clinical outcomes by using online cancer datasets [86]. In 2021, it was reported that carbamoyl–phosphate synthetase 1 (CPS1), a rate-limiting enzyme in urea cycle, was down-regulated in hepatocellular carcinoma. Reduction in CPS1 increased shunting of glutamine to CAD, making it easier for the de novo pyrimidine pathway to participate in cancer cell proliferation [87]. Additionally, the reduction of another enzyme in cancers, argininosuccinate synthase (ASS1), as well as CPS1, elevates cytosolic aspartate levels, thus providing substrates to CAD by facilitating pyrimidine synthesis [88]. This is to say that cancer cells hijack and redistribute existing metabolic pathways for cancer progression.

To our knowledge, CAD itself is found in a variety of cancer subtypes, consistent with a role for CAD in tumorigenesis. Most cancer occurs with hyperactivation of CAD, as well as the upstream, mTORC1 [51]. Besides, C-Myc, a hallmark of human cancers, also up-regulates CAD expression by binding with CAD E-box sequence in cancer cells [35,89]. Rapamycin-based mTOR inhibitors have been introduced into several clinical trials in the past couple of years. However, two issues have emerged on the clinical treatment: (1) S6K-based negative-feedback with enhanced PI3K-AKT activation, (2) upregulated macropinocytosis providing cancer cell amino acids [90]. CAD, as a downstream of mTORC1, has proved that its activity is a prerequisite for cancer cells to synthesize nucleic acids, and perturbation of nucleic acid biosynthesis can result in cancer cell death [91]. As shown by accumulating evidence, the need of normal cells for pyrimidine nucleotides is largely covered by the reutilization and recycling from the process of cell turnover or from dietary pyrimidine nucleotides [92,93]. Nevertheless, it is insufficient to cover the high demand in highly proliferating cancer cells for their own malignant benefit (Figure 4) [94,95,96]. CAD hyperactivation is a common event in tumors and mostly linked with tumor in two respects, i.e., metabolic programming and chemoresistance. Thus, CAD has recently emerged as a putative clinical target. Wang et al. found that glioblastoma stem cells up-regulate the de novo pyrimidine synthesis pathway and portend poor prognosis of patients with glioblastoma. Targeting CAD or the critical downstream enzyme DHODH via alleviating carbon influx through pyrimidine synthesis inhibited cancer cell survival, self-renewal [82,97]. For patients with hepatocellular carcinoma, high expression of CAD also predicted a shorter overall survival. It has been found that nitrogen can be redirected toward CAD and increase pyrimidine synthesis upon urea cycle dysregulation in cancer cells [87]. Collectively, inhibition of CAD or any other enzymes involved in de novo pyrimidine synthesis to deplete pyrimidine deoxyribonucleoside and ribonucleotide triphosphate pools has been proposed as a strategy for tumor treatment [98]. Presumably, inhibitors of pyrimidine synthesis enzymes might be potential drugs for cancer treatment (Table 1). However, there is still a big challenge in inhibitors targeting pyrimidine synthesis enzymes. A previous study revealed that acivicin could inhibit CPSII activity and attenuate the production of carbamyl phosphate [99], PALA and 4,5-dicarboxy-2-ketopentyl could suppress ATC activity [98], DUP-785 could inhibit DHODH, with a broad spectrum of antitumor activity, reflecting not only inmurine L1210 leukemia and colon carcinoma Colon 38 but also in human xenografts [100]. Presumably, inhibitors of pyrimidine synthesis enzymes might be a potential target for cancer cure. However, lesser inhibitors can be used in clinic. For example, PALA was tested and terminated in a phase II trial because of its low efficacy and damage to normal cells [35]. Another inhibitor, brequinar, targeting DHODH, unluckily, failed in clinical trials although it has been proven to be efficacious in animal models [101,102,103]. Some explanations for failure of CAD or single component inhibitors in clinical trials may be: (1) DNA damage to normal cells (e.g., PALA); (2) inhibitor efficacy struggles to last, for example, uridine nucleotide pools recover rapidly after Brequinar depletion, although Brequinar has the high efficiency to inhibit UTP; (3) other pathways or genes are altered when treated with de novo pyrimidine pathway-related inhibitors. It has been reported that DHODH is a remarkably frequent target for activators of p53, which gets a synergetic effect on efficacy of against cancer cells. Therefore, combination of DHODH inhibitors and p53 activator has been identified to promote cancer cel killing [104]. As well as brequinar targeting DHODH, other advanced DHODH inhibitors are also being tested in pre-clinics. Thus, there might be a specifically CAD or single component inhibitor in the future, but there is still a long way to go.

### 4.2. CAD and Inherited Metabolic Diseases

CAD encodes a multifunctional enzyme complex and plays pivotal roles in a variety of biological functions. Thereby, partial or total loss of function of CAD may cause severe congenital metabolic disorders [105]. Abnormal changes of CAD, such as genetic mutations, result in severe metabolic disease and even have fatal outcomes. This was first reported in 2015 that an individual who is compound heterozygous for mutations in CAD CPSA domain (c.1843-1G > A) and CAD ATC domain (c.6071G > A) and manifests as lower RNA and DNA through the de novo synthesis pathway and UDP-activated glycosylation level [41]. Subsequently, four similar cases were found in 2017 from three unrelated families. It has been reported that biallelic CAD mutations can be lethal. In these four cases, children had global developmental delay, epileptic encephalopathy, and anaemia with anisopoikilocytosis [14]. Collectively, CAD variants and clinical summary are listed in Table 2. Except for CAD, mutations of any enzymes involved in the de novo pyrimidine biosynthesis pathway also lead to metabolic diseases. For example, UMPS mutation at c.254T > C and c.1027 C > A causes UMPS-deficiency and results in hereditary oroticaciduria [106,107]. DHODH mutations are associated with Millersyndrome [108,109].

### 4.3. CAD and Immunity

A nucleoside pool in the cellular environment benefits not only host cells but also virus and bacteria. Therefore, pyrimidine biosynthesis enzymes are regarded as attractive targets for antiviral or antibiosis drug development. Brequinar, as mentioned earlier, a DHODH inhibitor, combined with another specific inhibitors of DHODH, is identified to inhibit virus replication and trigger the transcription of antiviral interferon-stimulated genes [101]. UDP-Glucose is also involved in modulating immune responses by activating the UDP-Glucose receptor. Sesma et al. unmasked that UDP-Glucose enhanced neutrophil lung recruitment via the P2Y_14_ receptor, but was abrogated when the receptor was a retardant [110]. Additionally, UDP–glucose stimulated IL-8 production via the P2Y_14_ receptor in human endometrial epithelial cells [111]. Furthermore, NOD2, a member of nucleotide-binding oligomerization domain (NOD)-like families (NLR family), are pattern recognition receptors that sense conserved microbial associated molecular patterns and trigger a signaling cascade leading to secretion of pro-inflammatory cytokines, chemokines, antimicrobial peptides [112,113]. Interestingly, it has been revealed that NOD2 could combine with CAD at its CPSase domain to modulate its own function in recent research spots [114,115,116,117,118,119,120,121,122]. NOD2 is involved in the development of Crohn’s disease (CD). Interestingly, CAD has been identified as a potential therapeutic target of CD by inhibiting NOD2-dependent activation [123]. Inhibition of pyrimidine synthesis via PALA (targeting CAD ATCase domain) also enhanced cellular secretion of the antimicrobial peptides in humans. It is worth noting that pyrimidine synthesis inhibitions have no direct anti-bacteria effect. Thus, it is proved that CAD is a negative regulator of NOD2 and has roles in antibacterial and antiviral reactions [124].

### 4.4. CAD and Neurological Disorders

The de novo pyrimidine pathway plays an important role in maintaining brain–body balance, neuron differentiation, and function of nerve mitochondria [32,125,126]. Though pyrimidines in neural tissues mainly derives from extraneural preformed pyrimidines sources, i.e., blood and liver [127,128], enzymes of the de novo pyrimidine synthesis pathway still play critical roles in mammalian brain development and health [32] (Figure 5). As introduced earlier, differentiated cells mainly depend on the pyrimidine salvage synthesis pathway; however, some neuronal differentiation processes, such as formation of axons and dendrites and maintenance of the neuron’s surface extension, still require a continuous membrane synthesis [126]. CAD (p. Arg 2024 Gln) mutation with loss function of carbamoyl–phosphate binding, causes developmental delays and epileptic encephalopathy, which implicates that CAD deficiency is linked tightly with neurometabolic disorders. Fortunately, oral treatment with uridine, which compensates for the de novo pyrimidines pathway, alleviates the symptoms of these diseases. Additionally, de novo pyrimidines pathway also plays a vital role in the pathogenesis of neurodegenerative diseases, such as Parkinson’s disease and Alzheimer’s disease (AD) [14,41]. These diseases are characterized by obvious damage in synapses and a continuous loss of neurons and synaptic proteins [129]. According to the ‘mitochondrial cascade hypothesis’, AD mostly exhibits a defect in the oxidative phosphorylation (OXPHOS) system. Dysfunction of this system affects the activity of DHODH and de novo pyrimidine biosynthesis, thereby compromising neuronal membrane formation and synapses production. Additionally, it has been identified that a new pyrimidine derivative, 4-benzsulfamide offers high therapeutic efficacy and low systemic toxicity in chronic traumatic encephalopathy by regulating mitochondrial function [130]. Scientists from another lab also revealed other pyrimidine derivatives as a potential candidate in AD treatment [131]. Another finding is that P2Y2, a pyrimidine-sensitive receptor, is reduced in AD patients in the parietal cortex, which is correlated with both neuropathologic scores and markers of synapse loss [132]. Luckily, nucleosides, like cytidine and uridine, can rescue the symptoms and take part in the formation of synaptic protein and membrane phosphatide [126,133,134]. The mechanisms by which supplemental uridine or pyrimidine derivatives increase phosphatide synthesis probably include two processes: (1) promoting uridine convert into endogenous CDP-choline, (2) activating pyrimidine-sensitive receptors (P2Y family) by UTP or pyrimidine derivatives so as to affect neurite outgrowth [134].

## 5. Perspectives and Conclusions

It is now clear that CAD plays a central role in de novo pyrimidine biosynthesis and nucleic acid synthesis, among others. In just the last few years, many new ideas about the function, structure, and rules of CAD have been clarified. Furthermore, extensive studies have enhanced our understanding in the relationship between CAD dysfunction and related diseases. However, though many new insights into CAD function and regulation have been elucidated, a systematic understanding of CAD is still not available, that is, while research on CAD continues, how to unmask the mechanism on why and how it exerts in the clinic and find a therapeutic target is still a question. For inherited metabolic diseases, finding the pathogenic mechanism and fulfilling genetic testing on next generation is pivotal. Genetic testing techniques should be further applied to prevent this rare neurometabolic disease from affecting human beings. Currently, the development of more sensitive screening methods is imperative for improved detection and treatment of disorders related to CAD.

Although many inputs to CAD inhibitors have been successfully used to suppress particular cancer types, there is still much work to deal with. For example, in spite of the important functions of CAD inhibitors on cancer, nearly most of them (AICAr, leucovorin, N-phosphonoacetyl-l-aspartic acid, etc.) have failed in clinic evaluations. Besides, there are two conditions that CAD inhibitors face: non-specific effects and genome toxicity. Further explorations into the regulation and functional aspects of CAD will provide deeper and more fundamental insights into the biological roles of CAD and its therapeutic potential for various diseases. Thus, more work should be focused on finding the targets of tumors, as well as CNS diseases, in CAD per se and its interacted factors. In detail, post-translational modifications of CAD in disorders require more in-depth research. Further studies should focus on identifying CAD-interacting enzymes or proteins, and exploring how these interactions regulate the activity, stability, or subcellular localization of CAD. Specifically, some proteins such as myc and Rheb have been reported, but lots of other proteins that can regulate CAD also need to be unmasked. These studies not only facilitate the understanding of the targets of disease development, but also benefit diagnostic evaluation and therapeutic strategies in clinics.

## Figures and Tables

**Figure 1 ijms-22-10253-f001:**
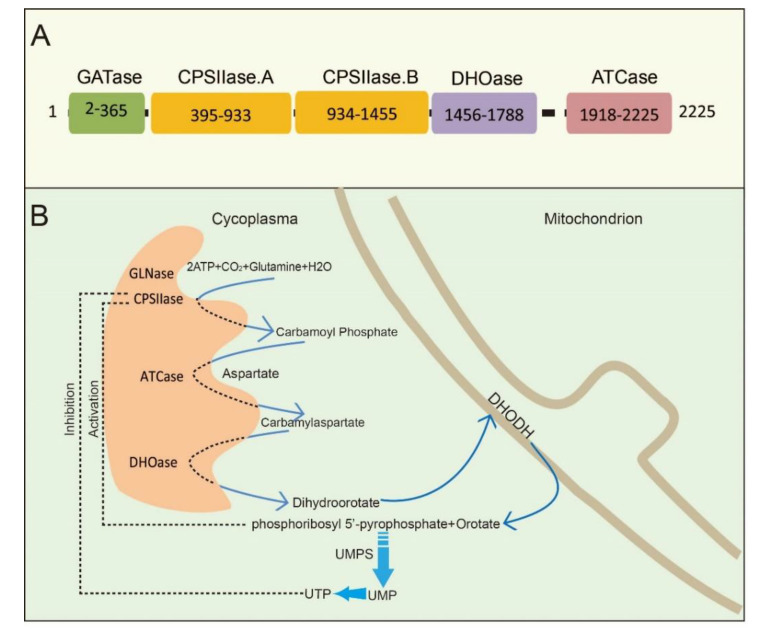
The schematic illustration of CAD and CAD-mediated de novo synthesis of pyrimidine nucleotides. (**A**) CAD and its domains are shown with the amino acid numbers. (**B**) The process of de novo synthesis of pyrimidine nucleotides. The first three steps are catalyzed by a trifunctional, cytoplasmic enzyme known as CAD, an acronym derived from the names of the three activities in this protein: carbamoyl phosphate synthetase, aspartate transcarbamylase, and dihydroorotase. The fourth enzyme is a mitochondrial enzyme, dihydroorotate dehydrogenase (DHODH). Phosphoribosyl 5′-pyrophosphate (PRPP) and orotate are catalyzed by UMP synthetase (UMPS) to produce UMP.

**Figure 2 ijms-22-10253-f002:**
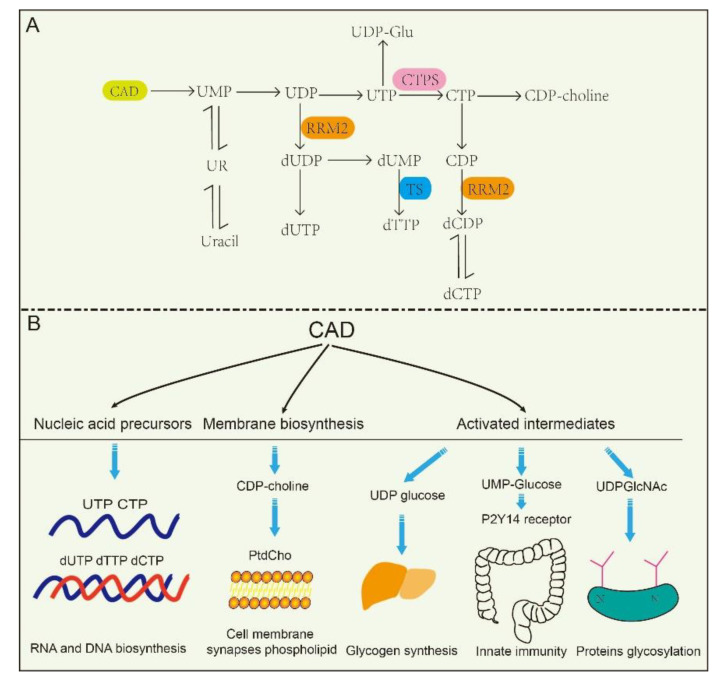
The metabolic pathways and related functions of pyrimidine nucleotides. (**A**): CAD participates in de novo pyrimidine nucleotides synthesis and metabolic pathways in mammalian cells. (**B**): CAD plays a role in the synthesis of nucleic acid precursors, activation intermediates and membrane. UMP, uridine monophosphate. UDP, uridine diphosphate. UTP, uridine triphosphate. CTPS, cytidine triphosphate synthetase. CTP, cytidine triphosphate. UR, uridine. dUDP, deoxy-uridine diphosphate. dUTP, deoxy-uridine triphosphate. dUMP, deoxy-uridine monophosphate. dTTP, deoxy-thymidine triphosphate. CDP, cytidine diphosphate. dCDP, deoxy-cytidine diphosphate. dCTP, deoxy-cytidine triphosphate. TS, thymidylate synthase. CDP-choline, cytidine diphosphate-choline. UDP-Glu, uridine diphosphate-glucose. RRM2, ribonucleotide reductase M2.

**Figure 3 ijms-22-10253-f003:**
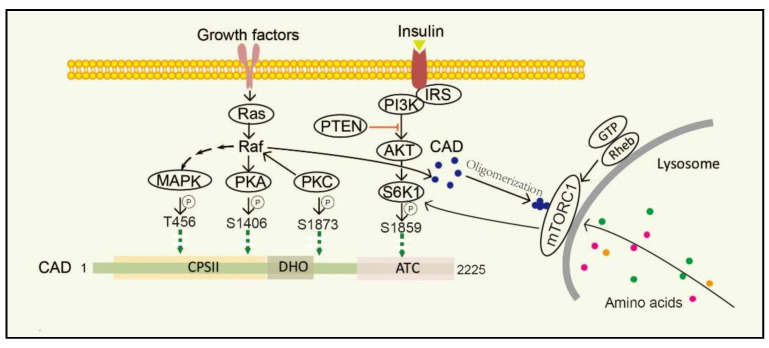
Diagram of regulation and phosphorylation sites in CAD. CAD oligomerization is promoted under amino acids stimulation and more easily anchored onto the surface of lysosome. Lysosome provides CAD a physical environment and gives it better access to glutamine, which is needed for de novo pyrimidine biosynthesis. Growth factor activates MAPK, PKA, and PKC phosphorylation sites in the CAD protein. S6K1 phosphorylates CAD via the mTORC1 pathway activated by animo acids or the PI3K/AKT pathway activated by insulin. CPSII, the carbamoyl phosphate synthetase domains. DHO, the dehydrooratase domain. ATC, the ATCase domain. EGF, epidermal growth factor. MAPK, mitogen-activated protein kinase. PKA, protein kinase A. PKC, protein kinase C. mTORC1, mammalian target of rapamycin complex 1. S6K1, S6 kinase 1.

**Figure 4 ijms-22-10253-f004:**
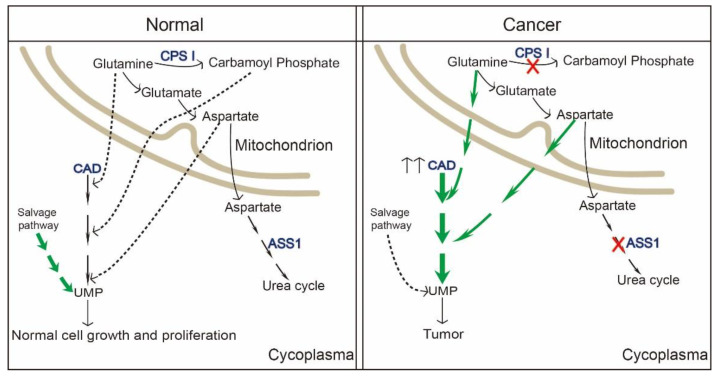
The role of CAD in cancer. Upstream of CAD and modification of the pyrimidine nucleotide metabolic pathway result in a wide variety of cancers. In normal cells, the need for pyrimidine is largely covered by the salvage pathway. In cancer cells, adaptive transformation from the salvage pathway into the de novo pyrimidine synthesis pathway benefits their high nucleotide requirement for malignant proliferation. Arrows with a solid or dotted line represent dominant or non-dominant metabolic flow in the pyrimidine nucleotide synthesis pathway. CPSI, carbamylphosphatesynthetase I. ASS1, argininosuccinate synthase.

**Figure 5 ijms-22-10253-f005:**
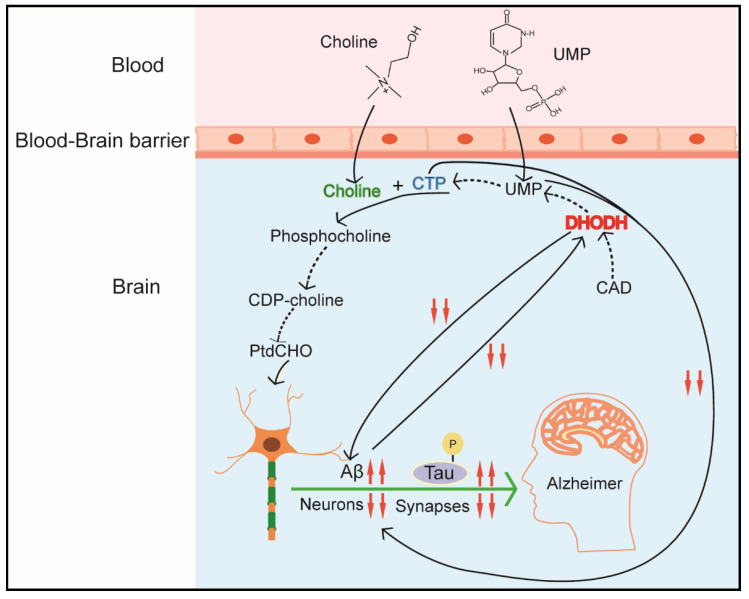
CAD in brain function. UMP, uridine monophosphate. CTP, cytidine triphosphate. CDP-choline, cytidine diphosphate-choline. PtdCHO, phosphatidylcholine. DHODH, dihydroorotate dehydrogenase.

**Table 1 ijms-22-10253-t001:** Relationship between CAD and clinical therapy of tumor.

Tumor or Model Type	Inhibitors	Target	Clinical Traits (+) or Not (−)	References
Colorectal cancer	N-phosphonoacetyl-l-aspartic acid leucovorin	ATCase	−	[135,136,137]
Lymphoma Melanoma	N-(phosphonacetyl)-L-aspartate (PALA)	ATCase	+, terminated, Clinical phase II	[138,139]
Myeloid leukemia	ASLAN003	DHODH	+, processing in clinical phase II	[35]
Myeloid malignancies	BAY 2402234	DHODH	+, terminated, clinical phase II	[140,141]
Myeloma cell lines	AICAr	UMPS	−	[142]
central nervous system disorder (multiple sclerosis)	Teriflunomide	DHODH	+	[82]

**Table 2 ijms-22-10253-t002:** CAD Variants and Clinical Summary.

Effect on Function of CAD or Related Clinical Phenotypes	Variations in CAD	Mechanism	Location	References
CAD loss catalytic activity	p.His1471Ala/p. His1473Ala	Active site zinc mutants, no zinc-binding	DHOase	[19,143]
	p.His1590Asn/p. His1614Asn	Alter the coordination of Zn-βcoordinating residue		
	p.Cys1613Ser	Alter the coordination of Zn-γ coordinating residue		
	P.Thr1562Ala/p. Thr1563Ala	Disturb and make the structure of CAD unstable		
	p.Asp1686Asn	Affect the coordination of Zn-α		
Activity of CAD decreased nearly 100-fold compared to wild-type, CAD catalytic activity↓	p.Glu1637Thr	Alter the coordination of Zn-γ coordinating residue	DHOase	[19]
11.5% of wild-type catalytic activity of DHOase, CAD catalytic activity↓	p.His1642Asn	Bind similar amounts of zinc compared to wild type, but a 3-fold increase of Km	DHOase	[143]
2.9% of wild-type catalytic activity of DHOase, CAD catalytic activity↓	p.His1690Asn	A 9-fold increase of Km, pH dependence in both the degradative and the biosynthetic decreased	DHOase	[143]
CAD catalytic activity↓, PMA-induced Thr-456 phosphorylation↓	p.Ser1873Ala	AlterPKC phosphorylation site	-	[13]
CAD mutations, neurometabolic disorder (Seizures, developmental delay, etc.)	c.98T > G	Main inducement of epileptic encephalopathy	GATase	[14,144,145]
	c.1843-3C > T	Affect the splice acceptor site of intron 12	CPSIIase.B	
	c.5365C > T	Polypeptide missed at the last 438 of 2225 amino acids of CAD protein	DHOase	
Biallelic mutations in CAD, de novo pyrimidine biosynthesis↓, glycosylation precursors↓	c.1843-1G > A	In-frame deletion of exon 13	CPSIIase.B	[41]
	c.6071G > A	Carbamoyl-phosphate binding↓	DHOase	

PKC, Protein kinase C; ↓,Decrease.

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
