# Peer review of "Pyrimidine Biosynthetic Enzyme CAD: Its Function, Regulation, and Diagnostic Potential"

_ijms, 2021, doi:10.3390/ijms221910253_

Round 1

Reviewer 1 Report

Comments on Manuscript ID: ijms-1361562: “Pyrimidine biosynthetic enzyme CAD as a therapeutic target: its function, regulation, and diagnostic potential" by Li et al.

The authors summarized the CAD pathway of pyrimidine biosynthesis.

The following things need to be addressed in MS:

  1. The title: CAD as a therapeutic target is unclear from the MS becoz it is not discussed in the MS how CAD which is actually the acronym of three enzymes, can be targeted in different metabolic disease conditions without any side effect on normal cells? Moreover, the MS discusses the CAD pathway in general-so the title should be changed.
  2. Line 267-286: Author tried to discuss how people targeted different enzymes (single) using different inhibitors in clinical trials, but most of them failed or terminated-the reasoning for failure is not discussed in the MS, even not hypothesized, what might be the reason of targeting different enzymes of CAD failed? That might also get connected to the first comment.
  3. Figure 4 is not properly described in the text and figure legend, especially the different signature (arrows) meaning.
  4. Section 10: Perspectives and conclusion-first para, line 371-384 is a general conclusion not matching with the MS-for example-line 382 “disease prevention is much more important than treatment”-this a general sentence-whole para should be rewritten in context to the CAD signaling.
  5. The MS needs proof-reading-few examples here: typos Line 272, line 275, table 1-column 1-myelomacell lines.

The paper is technically sound and well written. comments above- will make the MS better.

Author Response

We thank you very much for the critical comments and helpful suggestions. We have taken all these comments and suggestions into account, and they have improved our manuscript considerably.

Reviewer: Comments on Manuscript ID: ijms-1361562: “Pyrimidine biosynthetic enzyme CAD as a therapeutic target: its function, regulation, and diagnostic potential" by Li et al. The authors summarized the CAD pathway of pyrimidine biosynthesis.

Response: We agree with you.

Reviewer: 1. The title: CAD as a therapeutic target is unclear from the MS becoz it is not discussed in the MS how CAD which is actually the acronym of three enzymes, can be targeted in different metabolic disease conditions without any side effect on normal cells? Moreover, the MS discusses the CAD pathway in general-so the title should be changed.

Response: According to your valuable suggestion, we have changed the title “Pyrimidine biosynthetic enzyme CAD as a therapeutic target: its function, regulation, and diagnostic potential” to “Pyrimidine biosynthetic enzyme CAD: its function, regulation, and diagnostic potential”.

Reviewer: 2. Line 267-286: Author tried to discuss how people targeted different enzymes (single) using different inhibitors in clinical trials, but most of them failed or terminated-the reasoning for failure is not discussed in the MS, even not hypothesized, what might be the reason of targeting different enzymes of CAD failed? That might also get connected to the first comment.

Response: We have discussed the reasons for failure of the inhibitors targeting different enzymes of CAD.

Reviewer: 3. Figure 4 is not properly described in the text and figure legend, especially the different signature (arrows) meaning.

Response: We have revised the description in the text and figure legend of Figure 4.

Reviewer: 4. Section 10: Perspectives and conclusion-first para, line 371-384 is a general conclusion not matching with the MS-for example-line 382 “disease prevention is much more important than treatment”-this a general sentence-whole para should be rewritten in context to the CAD signaling.

Response: We have revised the paragraph.

Reviewer: 5. The MS needs proof-reading-few examples here: typos Line 272, line 275, table 1-column 1-myelomacell lines.

Response: We have corrected these errors.

Reviewer 2 Report

In this review, the authors first discussed the structure, function and regulation of CAD. Then they reviewed the role of CAD in human diseases including cancer, neurological disorders, and inherited metabolic diseases. Overall, this review includes most aspects of CAD. In addition, the figures and tables clearly illustrated the related functions or signaling pathways. However, the review was not well organized and presented.

1 2.3 The regulation of CAD, 3 MAPK/CAMPP-dependent PKA/PKC pathway, 4. PI3K-AKT-mTORC1-S6K1 pathway should be under 3 The regulation of CAD which includes these three parts. 3.1 The regulation of CAD; 3.2 MAPK/CAMPP-dependent PKA/PKC pathway; 3.3 PI3K-AKT-mTORC1-S6K1 pathway.

2 The part5, 6, 7, 8, 9 should be under 4 CAD in health and disease. 4.1 CAD and tumor; 4.2 CAD and inherited metabolic diseases; 4.3 CAD and immunity; 4.4 CAD and neurological disorders.

3 The part of mTORC1 in cancer should either combine with part  4 PI3K-AKT-mTORC1-S6K1 pathway or delete the redundant part.

Author Response

We thank you very much for the critical comments and helpful suggestions. We have taken all these comments and suggestions into account, and they have improved our manuscript considerably.

Reviewer: In this review, the authors first discussed the structure, function and regulation of CAD. Then they reviewed the role of CAD in human diseases including cancer, neurological disorders, and inherited metabolic diseases. Overall, this review includes most aspects of CAD. In addition, the figures and tables clearly illustrated the related functions or signaling pathways. However, the review was not well organized and presented.

Response: We agree with you.

Reviewer: 1. 2.3 the regulation of CAD, 3 MAPK/CAMPP-dependent PKA/PKC pathway, 4. PI3K-AKT-mTORC1-S6K1 pathway should be under 3 The regulation of CAD which includes these three parts. 3.1 The regulation of CAD; 3.2 MAPK/CAMPP-dependent PKA/PKC pathway; 3.3 PI3K-AKT-mTORC1-S6K1 pathway.

Response: We have revised the manuscript according to your valuable suggestion.

Reviewer: 2. The part5, 6, 7, 8, 9 should be under 4 CAD in health and disease. 4.1 CAD and tumor; 4.2 CAD and inherited metabolic diseases; 4.3 CAD and immunity; 4.4 CAD and neurological disorders.

Response: We have revised the manuscript according to your valuable suggestion.

Reviewer: 3. The part of mTORC1 in cancer should either combine with part 4 PI3K-AKT-mTORC1-S6K1 pathway or delete the redundant part.

Response: We have revised the manuscript according to your valuable suggestion.